# The PKA-p38MAPK-NFAT5-Organic Osmolytes Pathway in Duchenne Muscular Dystrophy: From Essential Player in Osmotic Homeostasis, Inflammation and Skeletal Muscle Regeneration to Therapeutic Target

**DOI:** 10.3390/biomedicines9040350

**Published:** 2021-03-30

**Authors:** Sandrine Herbelet, Caroline Merckx, Boel De Paepe

**Affiliations:** 1Department of Neurology, Ghent University and Ghent University Hospital, C. Heymanslaan 10, 9000 Ghent, Belgium; sandrine.herbelet@ugent.be (S.H.); caroline.merckx@ugent.be (C.M.); 2Neuromuscular Reference Center, Ghent University Hospital, C. Heymanslaan 10, 9000 Ghent, Belgium

**Keywords:** Duchenne muscular dystrophy, PKA-p38MAPK-NFAT5-organic osmolytes pathway, therapy

## Abstract

In Duchenne muscular dystrophy (DMD), the absence of dystrophin from the dystrophin-associated protein complex (DAPC) causes muscle membrane instability, which leads to myofiber necrosis, hampered regeneration, and chronic inflammation. The resulting disabled DAPC-associated cellular pathways have been described both at the molecular and the therapeutical level, with the Toll-like receptor nuclear factor kappa-light-chain-enhancer of activated B cells pathway (NF-ƘB), Janus kinase/signal transducer and activator of transcription proteins, and the transforming growth factor-β pathways receiving the most attention. In this review, we specifically focus on the protein kinase A/ mitogen-activated protein kinase/nuclear factor of activated T-cells 5/organic osmolytes (PKA-p38MAPK-NFAT5-organic osmolytes) pathway. This pathway plays an important role in osmotic homeostasis essential to normal cell physiology via its regulation of the influx/efflux of organic osmolytes. Besides, NFAT5 plays an essential role in cell survival under hyperosmolar conditions, in skeletal muscle regeneration, and in tissue inflammation, closely interacting with the master regulator of inflammation NF-ƘB. We describe the involvement of the PKA-p38MAPK-NFAT5-organic osmolytes pathway in DMD pathophysiology and provide a clear overview of which therapeutic molecules could be of potential benefit to DMD patients. We conclude that modulation of the PKA-p38MAPK-NFAT5-organic osmolytes pathway could be developed as supportive treatment for DMD in conjunction with genetic therapy.

## 1. Introduction

Duchenne muscular dystrophy (DMD) is a severe X-linked disorder usually recognized in early childhood, characterized by progressive skeletal muscle weakness causing loss of ambulation by early adolescence. In addition to the skeletal muscle, the disorder also involves cardiac and respiratory muscles. DMD is caused by disruptive mutations in the *DMD* gene, resulting in the absence of its protein product dystrophin from muscle fiber membranes. Contraction damages the dystrophin-deficient muscle fibers, generating cycles of muscle fiber necrosis and regeneration that fail to restore the tissue, leading to fibrosis and fatty replacement. Absence of dystrophin distorts anchoring of the extracellular matrix to the myofiber cytoskeleton [1,2,3]. Dystrophin is part of a larger group of transmembrane proteins, called the dystrophin associated protein complex (DAPC), which has the ability to receive and transduce signals from in- and outside the myofiber and absorb shocks during muscle contraction [4,5]. This shock-absorbing effect of the DAPC is compromised in DMD patients and causes cell membrane instability, which start as soon as the fetus can move in utero. The DAPC is composed of dystrobrevins, dystroglycans, sarcoglycans, sarcospan, and syntrophin. The most widely used murine model for DMD are mdx mice that carry a nonsense mutation in exon 23 of the *dystrophin* gene. However, the disease phenotype of this murine model is milder than the human condition. While humans display progressive muscle weakness, mice develop active muscle tissue degeneration and regeneration at a young age, which later on continues at a slower pace. The DAPC is affected in DMD and mdx displaying both common and differential deficiencies (Table 1). In mdx mice, α- and β-dystroglycans are unstable, whereas expression of α-, β-, γ- and δ-sarcoglycans, sarcospan and α1-syntrophin is weak [6,7,8,9,10,11] and expression of β1-syntrophin and α-dystrobrevin is absent [12,13,14]. In DMD patients, the following DAPC proteins are less expressed to sometimes absent: α- dystroglycan, α-sarcoglycan, sarcospan, α1-syntrophin, α-dystrobrevin, and neuronal nitric oxide synthase (nNOS) [10,15,16,17,18,19,20,21,22]. The role of nNOS instability in mdx pathology is controversial. Indeed, normal nNOS activity reduced dystrophic symptoms in one mdx study, whereas mdx mice crossed with NOS-null mice showed no difference in muscle pathology when compared to mdx mice [23,24,25]. β-dystroglycan is still present in DMD skeletal muscle tissue [6] and expression studies of β1-syntrophin in DMD patients are not yet available. Both β-dystroglycan and syntrophin function as signaling proteins. More specifically, β-dystroglycan signals to Ras-related C3 botulinum toxin substrate 1 (Rac1) small guanosinetrifosfaat (GTP)ase and to mitogen-activated protein kinase (MAPK) through growth factor receptor-bound protein 2 (Grb2). Syntrophin organizes a signalplex linked to dystrophin and regulates signaling proteins such as voltage-gated sodium channels along with plasma membrane calcium pumps and nNOS. DAPC also interacts with calmodulin which is stimulated by calcium and in turn signals to calcineurin [5,26,27].

Hampered signal transduction and cellular pathways have been described in DMD with Toll-like receptor/tumor necrosis factor α (TNF-α)/interleukin 1β (IL-1β)/interleukin 6 (IL-6)-nuclear factor kappa-light-chain-enhancer of the activated B cell pathway (NF-ƘB), Janus kinase/signal transducer and activator of transcription proteins, and the transforming growth factor-β (TGF-β) pathways having been extensively studied and considered for therapeutic targeting [28,29]. Due to the important inflammatory aspect of the disease, glucocorticoids (GCs) are the drugs of choice in DMD with their major mode of action residing in the binding to NF-ƘB and MAPKs along with nuclear translocation of nuclear factor of activated T-cells (NFAT) [30,31]. The NFAT group consists of five transcription factors belonging to the larger ReI family which also encompasses NF-ƘB. NFATc1-4 are regulated by calcineurin, whereas NFAT5 is a non-calcineurin mediated transcription factor harboring similarities with both NFATc and NF-ƘB [32]. NFAT5 is a multifaceted protein, which tightly controls cell volume in order to remain inside the homeostatic range. It controls cell growth in embryogenic tissue, mediates inflammation and protects cells from oxidative stress and metabolic aberrations due for instance to excessive caloric intake. It therefore deserves the name of immunometabolic stress protein [33].

In DMD, NFAT5 could play a role in permanent extracellular matrix protein production by fibroblasts [34] and could serve as a binding site for glucocorticoid receptor (GR), possibly explaining its anti-proliferative role in fibrosis formation [35]. In healthy myoblasts, NFAT5 is an essential protein in cell migration during myogenesis, but in inflammatory disease pro-inflammatory cytokines hamper normal NFAT5 physiology [36,37]. In lymphocytes and renal medullary cells, hyperosmotic stimuli activate the guanine nucleotide exchange factor Brx, also named A-kinase anchor protein 13 (AKAP13), which belongs to the protein kinase A (PKA) family and is linked with Rac1. In turn, it activates p38αMAPK, glycogen synthase kinase 3 (GSK-3), and NFAT5 [38,39]. The pathway also involves osmolytes, which are protective solutes that safeguard cells from perturbations in volume and osmotic imbalance. Osmolytes are involved in normal skeletal muscle physiology and become dysregulated in DMD (Table 2). Dystrophin deficiency perturbs the muscle cell’s osmotic balance, probably due mostly to the passive efflux of osmolytes through the leaky plasma membranes. Activation of the osmolyte pathway in DMD may act to stabilize proteins and counteract tissue injury. 

In this review, we explore the PKA-p38MAPK-NFAT5-organic osmolytes pathway in DMD by providing an overview of the current knowledge and research gaps that need to be filled. Indeed, DAPC instability activates PKA. In turn, PKA activates both GSK3 and p38MAPK. The latter has an influence on NFAT5. Upon translocation to the nucleus, NFAT5 has the ability to activate genes coding for organic osmolytes carriers. A graphical representation of this overview is shown in Figure 1.

## 2. The PKA-p38MAPK-NFAT5- Pathway as a Therapeutic Target in Duchenne Muscular Dystrophy

### 2.1. Protein Kinase A

PKA regulates sugar, glycogen and lipid metabolism in the cell. In skeletal muscle tissue, PKA is in close contact with the DAPC, more specifically with α- and β- dystrobrevin [40]. In mdx mice and DMD patients, PKA is downregulated along with the muscle-specific A-kinase anchor protein (AKAP) named myospryn. PKA activity is reduced by 50% in mdx mice [41].

PKA upregulating molecules, urocortins (Ucns), have been successfully used in mdx mice. Ucns are neuropeptides involved in inflammatory responses, anxiety and renal physiology [42]. In mdx, Ucns were administered daily s.c. for two- to three-week-old mdx mice, resulting in improved muscle resistance to mechanical stress, increased skeletal muscle mass, reduced necrosis in the diaphragm and slow- and fast-twitch muscles, along with normalized calcium influx [43]. When combined with anti-interleukin-6 receptor antibody (xIL-6R), Ucns improved the impaired force in mdx diaphragms, along with muscle shortening and mechanical work production [44]. This combination also resulted in the recovery of respiratory function and pharyngeal dilator muscle force in mdx mice [45,46]. Both molecules have not been tested yet in DMD patients.

### 2.2. P38 Mitogen-Activated Protein Kinase

P38 MAPK is responsive to cytokines, UV-light, osmotic perturbations and heat shock and plays an important role in apoptosis. In skeletal muscle, constitutive p38 MAPK activation in satellite cells, as seen during ageing, impairs muscle regeneration [47,48]. Very recently, p38 MAPK has been shown to induce cell fusion in myotube formation. Besides, in aging and chronic inflammation, excessive p38 MAPK activation could disrupt skeletal muscle homeostasis, leading to muscle pathology and atrophy [49]. The exact role of p38 MAPK in mdx and DMD is not clear, but a protective role for p38 MAPK signaling is suggested [50]. In mdx mice, p38 MAPK is phosphorylated (*p*-p38) [51,52,53,54]. Two studies describe a decrease in *p*-p38 compared to wild type mice at rest [51,54], whereas one study mentions increased phosphorylation in mdx mice compared to wild type mice at rest [53]. The levels of *p*-p38 increase after treadmill exercise in mdx mice compared to wild type [52]. In one DMD patient, *p*-p38 was increased compared to normal individuals [55].

p38 MAPK-targeted therapeutic agents have been used in mdx mice with good results: free radical scavenger α-lipoic acid (ALA)/L-carnitine (L-CAR), p38 inhibitor SB203580 and the cyclooxygenase (COX)-2 inhibitor celecoxib [51,53,56]. Carnitine is a fundamental source of acetyl groups. As it acts by transporting long-chain fatty acids into the mitochondrial matrix, its bioavailability is directly related to the rate of ß-oxidation, which was found to be slow in DMD muscle [57], as were reduced carnitine levels [58]. With 95% of carnitine residing in skeletal muscle and since this tissue largely depends on fatty acids as an energy source, its use has been tried in DMD (50mg/kg/day) in an unsuccessful trial [59]. In Becker muscular dystrophy (BMD), a low level of carnitine was found and supplementation has been observed to be beneficial in some cases [60]. Carnitine might be helpful for cardiomyopathy but trials need more strict correlations and biomarkers. The use of steroid, carnitine and branched chain amino acids in DMD has been reviewed in 2015 [61]. ALA/L-CAR induced a reduction in the level of *p*-p38 in mdx diaphragms [51] and SB203580 increased the survival of mdx myofibers exposed to oxidative stress [53,56]. A four weeks, treatment of six-week-old mdx mice with celecoxib resulted in a fiber type switch from fast to slow phenotype and enhanced muscle fiber integrity and muscle strength. Increased utrophin A expression was noticed in the diaphragm, heart and tibialis anterior [56]. In mdx mice, utrophin reduces the dystrophic phenotype [62]. These three molecules have not yet been investigated in clinical trials involving DMD patients.

### 2.3. Glycogen Synthase Kinase 3 Beta

The serine/threonine protein kinase glycogen synthase kinase 3 (GSK-3) is ubiquitously expressed and exists in two isoforms, α and β. GSK-3β is a key regulator of a wide range of cellular functions, with a role in neuronal cell development and energy metabolism [63]. In skeletal muscle, it is involved in muscle regeneration [64,65], and its ablation results in accelerated regeneration after atrophy is induced by disuse [66]. Both in mdx mice and in a canine DMD model, the GSK-3 levels are elevated [67,68,69]. The exact role played by GSK-3 in DMD has not been fully investigated [69], but GCs are believed to interact with GSK-3β [70].

The selective GSK-3 peptide inhibitor, L803-mts, increased glucose transporter type 4 (GLUT4) expression in murine muscle tissue; the transporter is altered in DMD and seems to be involved in DMD insulin resistance [71,72]. No further data on GSK-3-targetted treatment is currently available for DMD.

### 2.4. Nuclear Factor of Activated T-Cells 5

NFAT5 or tonicity-responsive enhancer binding protein (TonEBP) is strongly present in DMD skeletal muscle tissue in nuclei of small myofibers and fibers with central nuclei [55], which could be linked to its role in muscle regeneration [36]. Staining with a Ser 1197-phosphorylated NFAT5 antibody shows this phosphorylation occurs around myonuclei, on the cell membrane and throughout the sarcoplasm in DMD tissue, especially in small fibers. In vascular smooth muscle cells, this phosphorylation appears to prevent NFAT5 from translocating to the nucleus and accumulates in the cytoplasm [73]. NFAT5 nuclear translocation is important during myogenesis [36]. This absence of NFAT5 translocation was observed in normal myoblasts exposed to pro-inflammatory cytokines, where NFAT5 formed aggregates in the sarcoplasm, without showing increased expression [37], and may also be present in DMD fibers, which are chronically exposed to inflammatory cytokines. This could lead to the decreased total NFAT5 protein expression in DMD patients we observe using immunoblotting. In the 15 muscle samples tested, we found constitutive expression in all 11 healthy controls but only in one of the four DMD samples a prominent band was present (Figure 2). NFAT5 has not been investigated in mdx mice so far. When proteins form pathogenic aggregates in the cytoplasm, one therapeutic option that could be considered are protein specific nanobodies [74], which will be discussed further on.

### 2.5. Factors Downstream of Nuclear Factor of Activated T-Cells 5

NFAT5 is known for activating genes that code for TNF-α, heat shock protein 70 (hsp70), and members of osmolyte pathways, in response to hypertonic stimuli [32,75,76]. Under hypertonic conditions, binding of NFAT5 to tonicity-responsive enhancer sequences located at the promotor region results in the increased gene transcription of osmolyte pathway members. In the next sections, individual osmolyte pathway members and their corresponding osmolytes are discussed in detail [77,78,79,80,81,82,83,84,85,86,87,88,89,90,91].

#### 2.5.1. Taurine Transporter

The taurine transporter (TauT or SLC6A6) is principally located on the cell membrane and transports taurine or 2-aminoethyl sulfonic acid from the extracellular environment into the cell. Taurine is an organic osmolyte and functions as an osmoregulator, but also plays a role in calcium homeostasis and exerts anti-inflammatory and antioxidant actions [92,93,94,95,96,97]. In DMD patients, protein levels of TauT are increased, particularly in the CD56+ small regenerating muscle fibers [55], whereas changes in taurine plasma and urinary levels, indicative of increased taurine excretion, have been reported [96,97]. In mdx mice, hampered regulation of the taurine pathway is manifested by a significant decrease of both taurine and its transporter in muscle tissue.

As a possible correlation exists between low taurine levels and muscle impairment [95,98,99,100,101,102], increasing intramuscular taurine levels could improve the mdx disease phenotype. Indeed, studies have found that taurine treatment in mdx mice improved histopathological features such as necrosis [102,103,104] and reduced markers of inflammation and oxidative damage [104,105,106]. Interestingly, Barker et al. reported the favorable effects of prenatal taurine administration (2.5% *w/v*) in young mdx mice (aged four weeks), whereas the beneficial effect of taurine on histopathological level was almost completely abolished in mice sacrificed at 10 weeks. While intramuscular taurine levels were increased (+25%) upon treatment in four-week-old mice in comparison to untreated mdx mice, the taurine levels were decreased (−22%) in adult treated mdx mice [107]. High taurine supplementation could induce a reduction in both the expression and activity of its transporter [94,108,109,110], which might explain lower intramuscular taurine levels. In the study by Barker et al., however, the protein expression of TauT was reported unchanged [107]. Taurine supplements might also exert beneficial effects on muscle strength. An increase in normalized fore limb muscle force measured by grip strength has been reported [103,111], but the in vitro measurement of specific muscle force was only able to show ameliorated muscle strength in young mdx mice [107], and not in adult mice [103,107]. Furthermore, a synergistic effect of taurine (1 g/kg) and α-methylprednisolone (1 mg/kg) on fore limb strength has been observed in mdx mice [112].

Taurine was able to improve histopathological features and muscle strength in the mdx mouse model and thus makes an interesting compound for the treatment of DMD. To our knowledge, no clinical trials have been conducted to evaluate the effect of taurine in DMD patients; however, the use of taurine in other conditions such as peripartum cardiomyopathy, obesity, etc. has been investigated (https://www.clinicaltrials.gov/, accessed on 10 February 2021).

#### 2.5.2. Betaine Gamma Aminobutyric Acid Transporter

The betaine gamma aminobutyric acid transporter (BGT or SLC6A12) transports betaine and to a lesser extent gamma aminobutyric acid (GABA) into cells. Betaine is an osmolyte that counters osmotic stress, and in addition acts as a methyl group donor in the conversion of homocysteine to dimethylglycine and methionine in the mitochondria of liver and kidney cells [80,81,82,83,93]. To our knowledge, betaine levels in the muscles of DMD patients have not been investigated; however, the increased expression of BGT in the small regenerating and atrophic muscle fibers of DMD patients has been reported, while staining was absent in control muscle tissues [93]. 

Betaine influences proliferation and differentiation in C2C12 cells [113,114]. The effect of betaine on proliferation and the differentiation in C2C12 myoblasts is abolished in the presence of 5-aza-2 deoxycytidine, a methylation inhibitor, which corroborates the hypothesis that betaine could regulate genes involved in skeletal muscle formation via epigenetic DNA methylation [113]. Furthermore, betaine might alter the distribution between fast- and slow-twitch muscle fibers in favor of slow-twitch myofibers. Slow-twitch fibers are more fatigue-resistant and contain more mitochondria, required for aerobic respiration. Betaine was reported to induce an upregulation of slow-twitch genes [113], presumably by the induction of NFATc1 and the subsequent attenuation of MyoD transcription. On the contrary, transcription of MyoD1 is stimulated in the breast muscle of partridge shank broiler chickens when supplemented with betaine [115]. Dose-dependent effects of betaine and/or differences in the metabolism between species might explain these conflicting results of betaine supplementation on MyoD transcription. In addition, betaine might exert beneficial effects on fatty tissue deposition. Betaine supplementation in mice significantly decreased the percentage of fat mass, whereas lean body mass of mice was increased [113]. Similar results were obtained in a mouse model for obesity. Betaine was able to reduce whole body fat, specifically perirenal, inguinal and gonadal fat, together with inflammatory stress-related genes such as IL-1, IL-6, CXCL13, etc. [116]. Betaine might affect the fatty acid synthesis pathway in muscle and was able to prevent a build-up of intramyocellular lipids. However, Wu et al. described increased lipid accumulation in vitro upon betaine supplementation [117]. These conflicting results might be attributed to differences between in vitro and in vivo models. To note, higher betaine intake in humans is associated with lower body fat [118,119].

Besides a reduction in fat mass, some researchers have reported increased skeletal muscle mass and endurance, both in mice and in humans, upon betaine supplementation [113,118,119,120]. Other beneficial effects of betaine supplementation on skeletal muscle have also been reported. For example, betaine exerted protective effects against skeletal muscle apoptosis caused by chronic alcohol overconsumption. Betaine treatment was able to significantly reduce cytochrome C-release, calpain activity, TNF-α, NF-ƘB, and creatine kinase levels [121]. Betaine supplementation might affect inflammation, skeletal muscle differentiation and lipid accumulation and it could therefore be worthwhile to explore using it as a supportive treatment option in DMD. Betaine is already approved by the Food and Drug Administration (FDA) as an orphan drug for the treatment of homocystinuria but has not yet been tried in DMD.

#### 2.5.3. Sodium/Myo-Inositol Co-transporter

The sodium/myo-inositol co-transporter (SLC5A3 or SMIT) transports myo-inositol (MI) into the cell. MI is a cyclic polyol and exerts numerous physiological functions. Besides its function as an osmolyte, protecting the cell against osmotic stress, MI can be phosphorylated to phosphatidylinositol 4,5-biphosphate (PIP2). PIP2 is a component of the plasma membrane and plays an important role in signal transduction. PIP2 can be further metabolized into diacylglycerol (DAG) or in inositol-(1,4,5) trisphosphate (IP3) that triggers the release of Ca^2+^ [122,123]. Protein levels of SMIT are readily detected in muscle tissue of DMD patients, in contrast to healthy subjects. Furthermore, immunofluorescent staining in muscle fibers of DMD patients revealed the presence of SMIT in a large part of muscle fibers and inflammatory cells (e.g., CD68+ macrophages and, to a lesser extent, CD206+ macrophages and T cells) [55,93].

The therapeutic effects of MI have been investigated primarily in the context of diabetes mellitus, a disease characterized by high glucose blood levels. Supplementation with MI in glucose-loaded mice caused a decrease in plasma glucose levels [124]. Similar beneficial effects of MI supplementation in obese insulin-resistant Rhesus monkeys on plasma glucose levels have been observed [125]. An MI-induced plasma glucose uptake in myotubes could, however, not be shown in vitro [126], which led to the hypothesis that the effect requires conversion of MI to D-chiro-inositol [124]. Rodriguez et al. reported hyperinsulinemia and insulin resistance in, respectively, 48.5% and 36% of DMD/Becker patients [72]. MI supplements have, however, not yet been tried in DMD patients.

#### 2.5.4. Aldose Reductase

Aldose reductase is an enzyme that catalyzes the reduction of glucose to sorbitol, the first reaction of the polyol pathway. Later, it was shown that its specificity was not limited to glucose and that the enzyme has a wide range preference for hydrophobic substrates such as steroids and lipid-derived/hydrophobic aldehydes [127,128,129,130,131,132,133]. Aldose reductase expression is high in skeletal muscle [134] and is increased even further in the muscle of DMD patients [55,93]. Similarly, an upregulation of aldose reductase is observed in the cardiac tissue of ageing mdx mice [135].

The role of sorbitol and aldose reductase is most extensively studied in models for diabetes. Elevation of sorbitol by 90% has been reported in the skeletal muscle of diabetic rats [136]. Supposedly, a decrease in the availability of adenosine triphosphate (ATP) molecules related to increased polyol pathway activity in diabetes might be associated with muscle dysfunction. Treatment with an aldose reductase inhibitor restored sorbitol levels to normal and had beneficial effects on relaxation and to a lesser extent on the contraction of skeletal muscle [136]. Contrastingly, insulin treatment can prevent rising sorbitol levels in diabetic rats but is incapable of ameliorating muscle strength in a significant manner [137,138]. Thus, the precise role of sorbitol on skeletal muscle dysfunction in diabetes is not entirely clear.

Furthermore, aldose reductase inhibition is also examined as a treatment for diabetes-induced vascular inflammation. Under hyperglycaemic conditions, NF-ƘB activity is enhanced in cultured vascular smooth muscle cells (VSMC) of rats. The research of Ramana et al. demonstrated that sorbinil, an aldose reductase inhibitor, abrogates high glucose mediated NF-ƘB activity presumably by interfering with the phosphorylation of IƘB-α and subsequent inhibition of NF-ƘB translocation to the nucleus. As expected, sorbinil prevents the expression of vascular cell adhesion molecule 1 (VCAM-1) and intracellular adhesion molecule 1 (ICAM-1), both target genes of NF-ƘB, in high glucose cultured VSMC’s. Interestingly, sorbinil is unable to inhibit osmotic stress-induced NF-ƘB activity. Apparently, hyperglycaemic and hyperosmotic conditions mediate NF-ƘB activity in, respectively, a protein kinase C (PKC)-dependent and PKC-independent manner. This difference in regulation might explain why aldose reductase inhibitors are able to inhibit high glucose-induced NF-ƘB activity, but not upon osmotic stress. Studies have reported an anti-inflammatory effect of aldose reductase inhibitors, which could be achieved by the modulation of NF-ƘB activity [139,140,141].

On the other hand, inhibition of aldose reductase aggravates tissue damage in an in vivo temporal artery severe combined immunodeficiency (SCID) mouse model for vasculitis. Aldose reductase plays a role in the detoxification of 4-hydroxynonenal (HNE), a biomarker for lipid peroxidation. Rittner et al. further investigated the relationship between aldose reductase and HNE in vasculitis. Sorbinil treatment in a mouse model for giant cell arteritis induced a twofold and threefold increase in, respectively, HNE adducts and apoptotic cells. Furthermore, stimulation of mononuclear cells with HNE induces the expression of aldose reductase. The latter might explain why an upregulation of aldose reductase is observed in T-cells, macrophages and smooth muscular cells surrounding arteritic lesions together with the presence of HNE in vivo [132]. These observations led to the hypothesis that aldose reductase interferes with lipid peroxidation and protects the arterial wall tissue against oxidative damage. Of note, clinical trials have been conducted to evaluate the effect of sorbinil for the treatment of diabetic retinopathy.

## 3. Discussion

GCs remain the mainstay of treatment today yet induce burdensome secondary effects in DMD patients [28]. The need persists for novel therapeutic avenues. Many researchers are focusing on gene therapy in order to restore functional dystrophin protein. Ataluren was the first drug approved by the FDA that addresses the underlying cause of DMD, namely the lack of functional dystrophin protein. Presumably, ataluren acts by stimulating the implementation of near-cognate tRNA’s selectively at the nonsense codon site, with the production of functional dystrophin protein as a result [142,143,144]. Eteplirsen has also received FDA approval for the treatment of DMD. Eteplirsen is a form of antisense oligonucleotide treatment that uses exon skipping in order to restore the reading frame and gives rise to an altered yet functional form of the dystrophin protein. Gene therapy is promising; however, there are some concerns regarding safety, efficacy, etc.

In this review, therapeutic molecules associated with the PKA-p38MAPK-NFAT5-organic osmolytes pathway are discussed in view of a supportive treatment for DMD. Ucns can upregulate PKA and may be worth considering in DMD clinical trials in conjunction with IL-6. Ucns belong to the corticotropin-releasing factor (CRF) and show both central and peripheral actions. In animals, it suppresses appetite, enhances locomotor activity, and has an anxiogenic effect along with a pro-inflammatory-inducing effect in the skin. In humans, Ucns lower the blood pressure [145,146]. More physiological studies in humans are needed before usage in DMD clinical studies. As described in a previous review, the p38 MAPK modulator (ALA)/L-car is an uninteresting molecule to consider despite its FDA approval and promising results in mdx mice. Indeed, it did not show any difference in the function of extremities in DMD patients [28]. An alternative molecule that alters p38 MAPK activity is Celecoxib. Celecoxib is used for the treatment of rheumatoid arthritis [147], is commercially available and may be worth considering in DMD clinical trials. However, the appearance of gastric ulcers should be kept in mind [147]. The selective inhibitor of p38 MAPK termed SB203580 and the GSK-3 peptide inhibitor L803-mts could be further explored in mdx mice.

NFAT5 is located in the cytoplasm of myoblasts, forming aggregates under pro-inflammatory conditions. Using anti-NFAT5 nanobodies could direct NFAT5 to the nuclei, which could enhance muscle regeneration. The latter would be achieved by myoblast-specific anti-NFAT5 nanobodies. In DMD fibroblasts, where NFAT5 is exclusively located in the nuclei, anti-NFAT5 would take NFAT5 to the cytoplasm, which might reduce fibrosis.

Downstream targets of NFAT5 involved in osmolyte pathways might also be considered as a supplementary therapeutic option. Aberrant calcium influx and the subsequent hyperosmolar environment, induction of organic osmolytes, TNF-α and hsp70 all play a role in the pathophysiology of DMD [53,148,149,150]. Besides countering osmotic stress, organic osmolytes exert a wide variety of effects that could be of importance in the treatment of DMD. The observed body fat-reducing capacities of betaine would be of benefit to DMD patients for standard GC therapy. In addition, the anti-inflammatory actions of osmolytes and their additive effects when combined with GCs would need to be elucidated further. Osmolytes are able to induce a structural conformational change in the AF1/Tau1 domain of the GR, which stimulates the interaction of AF1 with specific binding proteins such as TATA box binding protein (TBP), CREB binding protein (CBP) and steroid receptor coactivator-1 (SRC-1) [151,152]. Whether this fully explains a potential additive effect of osmolytes and GCs remains to be investigated. Of note, the synergistic effect of taurine and α-methylprednisolone in mdx mice reported by Cozzoli [112] could not be reproduced in the study of Barker [153]. This discrepancy might be attributed to differences in the analysis since Cozzoli et al. analysed the increase in muscle strength relative to a previous recording before treatment started and is thus less affected by individual differences.

Additionally, there are some potential pitfalls for osmolytes as a treatment. As mentioned earlier, studies have reported the inability of taurine treatment to increase muscle taurine levels in mdx mice; however, taurine treatment was able to significantly decrease inflammation [105,107]. Daily supplementation of taurine (5 g/day) in humans could not increase the taurine concentration in muscle either, but a potential anti-inflammatory effect was not investigated [154]. The latter might suggest that taurine levels are strictly regulated and supposedly, other organic osmolytes such as betaine, MI and sorbitol might also be subject to compensatory mechanisms of the metabolism. The inability to increase osmolyte levels upon supplementation might interfere with the long-term therapeutic effects. Furthermore, Barker et al. reported a negative side effect of high doses of taurine (16 g/kg/day). High taurine supplementation abrogates liver cysteine dioxygenase and cysteine sulfinate decarboxylase activity, which results in increased cysteine plasma levels. High levels of cysteine are considered toxic and might be associated with hampered growth, the latter is observed in taurine-treated mdx mice [105]. The overall effect of aldose reductase inhibitors in the treatment of dystrophinopathies are difficult to predict. As previously stated, the inhibition of aldose reductase might exert anti-inflammatory effects through the modulation of NF-ƘB activity [139,140,141]. On the other hand, aldose reductase inhibitors might abolish its antioxidant actions. This might result in higher HNE concentrations [132], which is often used as an outcome measure for the evaluation of oxidative stress in mdx mouse. Both positive and negative consequences of aldose reductase inhibitors have been reported.

## 4. Conclusions

Therapeutic targeting of the PKA-p38MAPK-NFAT5-organic osmolytes pathway could be considered as a supportive treatment for DMD in conjunction with genetic therapy or as putative molecules to be investigated in experimental DMD studies. Surely, novel therapeutic molecules should be evaluated first in vitro (DMD or mdx myotubes), subsequently in animal models (mdx mice, golden retriever muscular dystrophy dog) and may ultimately be tested in DMD/ BMD patients. Nowadays, carrying out clinical trials is difficult both due to the relative rarity of dystrophinopathies and family expectations, but are necessary in order to assess the effect of treatment on muscle performance, cardiomyopathy and osteoporosis in DMD and BMD patients [60].

Several research paths could be further explored such as expression studies of β1-syntrophin in DMD patients and the effect of celecoxib and the GSK-3 peptide inhibitor L803-mts. Osmolytes also make interesting compounds for the treatment of DMD due to their compatibility with GCs, anti-inflammatory and antioxidant actions. However, further investigation will be needed to find the optimal effective dose/treatment scheme that results in the highest therapeutic profit.

## Figures and Tables

**Figure 1 biomedicines-09-00350-f001:**
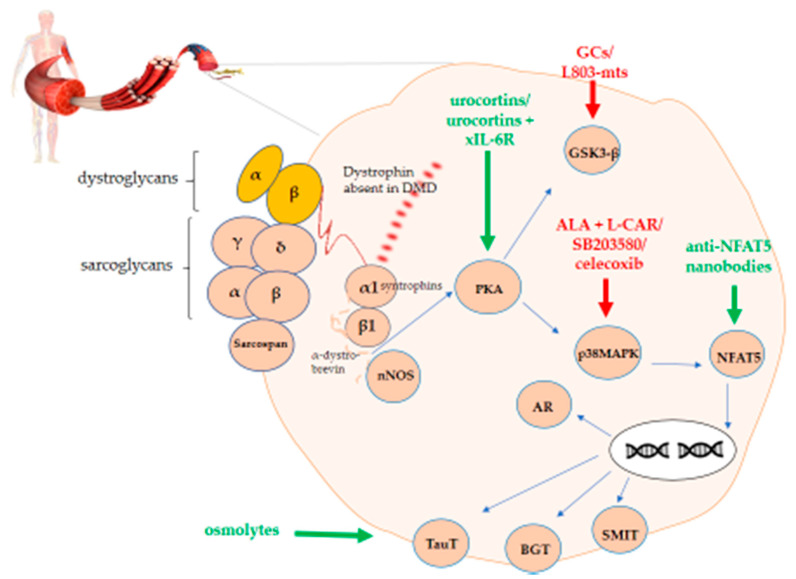
PKA-p38MAPK-NFAT5-organic osmolytes pathway in Duchenne muscular dystrophy (DMD). The figure shows the schematic representation of the protein kinase A/mitogen-activated protein kinase/nuclear factor of activated T-cells 5 (PKA-p38MAPK-NFAT5)-organic osmolytes pathway members representing potential therapeutic targets either to stimulate (green) or to inhibit (red) for treating DMD.

**Figure 2 biomedicines-09-00350-f002:**
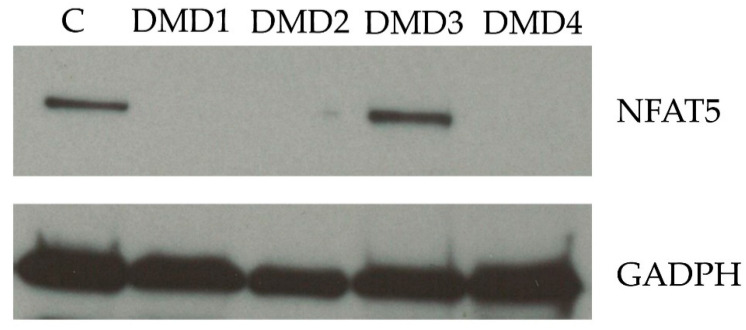
Nuclear factor of activated T-cells 5 (NFAT5) protein expression in skeletal muscle biopsies. Western blots are shown for skeletal biopsies from four patients affected with Duchenne muscular dystrophy (DMD) and from one representative healthy control (C). Total protein extracts were transferred to nitrocellulose membranes following electroblotting. Overnight incubation with 2 µg/mL mouse monoclonal IgG2a anti-NFAT5 (F-9, Santa Cruz) and 0.4 µg/mL anti-GAPDH (Sigma-Aldrich, St. Louis, MO, USA) (the latter used to correct for protein concentration differences between samples) was followed by immunoreaction detection with chemiluminescence (WesternBright™ Sirius, Advansta, Menlo Park, CA, USA) and Proxima 2650 (Isogen Life Science, De Meern, The Netherlands).

**Table 1 biomedicines-09-00350-t001:** Differential expression of the main dystrophin-associated protein complex (DAPC) components in Duchenne muscular dystrophy (DMD) and in the mdx mouse.

Main Components of the DAPC Complex	DMD Patients	mdx Mouse
α-dystroglycan	weak	unstable
β-dystroglycan		unstable
α-sarcoglycan	weak	weak
β-sarcoglycan		weak
δ-sarcoglycan		weak
γ-sarcoglycan		weak
sarcospan	weak	weak
dystrophin	absent	absent
α1-syntrophin	weak	weak
β1-syntrophin		absent
α-dystrobrevin	weak	absent

**Table 2 biomedicines-09-00350-t002:** The role of organic osmolytes in skeletal muscle physiology and in Duchenne muscular dystrophy (DMD) and its mouse model mdx.

Organic Osmolyte	Physiological Role	Expression in DMD Patients/mdx Mouse	Clinical Trials
Taurine	Osmoregulation, antioxidant, modulation of Ca^2+^ signaling, conjugation of bile acids	DMD: ↑ taurine excretion ↑ taurine trans-porter (TauT) mdx: ↓ muscle taurine ↓ TauT	Peripartum cardiomyopathy, thalassemia major, diabetes, etc.
Betaine	Osmoregulation, methyldonor	DMD: expression in small regenerative and atrophic fibers	Homocystinuria, non-alcoholic fatty liver disease, etc.
Myo-inositol (MI)	Osmoregulation, structural base for 2nd messengers (signal transduction, component of plasma membrane)	DMD: ↑ sodium/myo-inositol transporter (SMIT)	Polycystic ovary syndrome, bipolar disorders, etc.
Sorbitol	Osmoregulation	DMD: ↑ Aldose reductase mdx: ↑ Aldose reductase (cardiac muscle)	

## Data Availability

The data presented in this study is not publicly available, but available on request from the corresponding author.

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
