# Peer review of "The PKA-p38MAPK-NFAT5-Organic Osmolytes Pathway in Duchenne Muscular Dystrophy: From Essential Player in Osmotic Homeostasis, Inflammation and Skeletal Muscle Regeneration to Therapeutic Target"

_biomedicines, 2021, doi:10.3390/biomedicines9040350_

Round 1
Reviewer 1 Report
This review considers the involvement of PKA-p38MAPK-NFAT5-organic osmolite pathway in the possible potential benefit in DMD.Steroids remain so far the golden standard for DMD children and their use is the first line ,but they propose a potential supplementary role of alfa lipoic acid, Celecoxib, L-carnitine,taurine,betaine,myoinositol and sorbitol.
Carnitine is a fundamental source of acetyl groups. As it acts by transporting long‐ chain fatty acids into the mitochondrial matrix, its bioavailability is directly related to the rate of ß‐oxidation ,that was found slow in DMD muscle by Carrol,also carnitine level was decreased according toBerthiller et al.(1982). 95% of carnitine resides in skeletal muscle since this tissue largely depends on fatty acids as an energy source Its use has been tried in DMD by Escobar-Cedillo (50mg//kkg/day)in an unsuccessful trial, and a low level was found and also in BMD were in some cases was used (Angelini 2019),it might be helpful for cardiomyopathy but the trials need more strict correlations and biomarkers.The use of steroid ,carnitine, branched chain aminoacids in DMD is presented in Angelini, Tasca ,2015 Orphan Drugs: Research and Reviews with other drugs here mentioned. Celecoxib was found useful by Jasmin in mdx mice, butt here are reservation extending use of such drug in different species.
It is shown in Fig 1 the action of such compounds in mdx mice and DMD ,it might be relevant to fatigue in DMD see Acta Myol. 2012 ;31(1):9-15,that is an important feature, probably related to NOS defect.In the present review ,it should be made clear that mdx mice is not a representative animal model,because of it lacks several parameters including cycles of degeneration /regeneration in myofibers and has a poor fibrotic response, the figure should be more clear in this regard.
Figure 2 shows presence of NFAT5 protein by western blotting in one of four muscles of DMD ,NFAT5 could play a role in permanent regeneration but extracellular preoduction of matrix would inhibit this and could interfere as a possible binding site of glucocorticoid receptor(GR), differences in GR polymorphisms were found by Bonifati et al in steroid response in DMD children ,that is different according to his polymorphisms (J Neurol Neurosurg Psychiatry2006 Oct;77(10):1177-9)
Supportive treatment can be first tried in cell models, these therapeutic molecules could be tried in DMD myotubes or mdx myotubes(according to Terrill,2020 taurine supplementation decreases their necrosis), subsequently in animal models both in mdx and dog model, may be ultimately tested in DMD/ BMD patients,although carrying out trials is nowadays difficulty both for the relative rarity of dystrophinopaties and family expectations, are also worth trials in BMD (Acta Myol. 2019),in order to counteract both cardiomyopathy and osteoporosis.
The review presents a series of possible drugs to be tested as supplementary treatment in DMD, some were previously subjected to trials and their differential effects could be presented.
Author Response
Thank you kindly for your remarks, which we address point-by-point hereunder:
This review considers the involvement of PKA-p38MAPK-NFAT5-organic osmolite pathway in the possible potential benefit in DMD.Steroids remain so far the golden standard for DMD children and their use is the first line ,but they propose a potential supplementary role of alfa lipoic acid, Celecoxib, L-carnitine,taurine,betaine,myoinositol and sorbitol.
Carnitine is a fundamental source of acetyl groups. As it acts by transporting long‐ chain fatty acids into the mitochondrial matrix, its bioavailability is directly related to the rate of ß‐oxidation ,that was found slow in DMD muscle by Carrol,also carnitine level was decreased according toBerthiller et al.(1982). 95% of carnitine resides in skeletal muscle since this tissue largely depends on fatty acids as an energy source Its use has been tried in DMD by Escobar-Cedillo (50mg//kkg/day)in an unsuccessful trial, and a low level was found and also in BMD were in some cases was used (Angelini 2019),it might be helpful for cardiomyopathy but the trials need more strict correlations and biomarkers.The use of steroid ,carnitine, branched chain aminoacids in DMD is presented in Angelini, Tasca ,2015 Orphan Drugs: Research and Reviews with other drugs here mentioned. Celecoxib was found useful by Jasmin in mdx mice, but here are reservation extending use of such drug in different species.
We thank the reviewer for providing these valuable clarifications. The points raised have been addressed in section 2.2 of the revised manuscript.
It is shown in Fig 1 the action of such compounds in mdx mice and DMD ,it might be relevant to fatigue in DMD see Acta Myol. 2012 ;31(1):9-15,that is an important feature, probably related to NOS defect.In the present review ,it should be made clear that mdx mice is not a representative animal model,because of it lacks several parameters including cycles of degeneration /regeneration in myofibers and has a poor fibrotic response, the figure should be more clear in this regard.
The reviewer remarks on an important limitation of the mdx mouse as a model for DMD. Indeed, although the underlying genetic defect, which results in the lack of dystrophin protein, mirrors human disease, the disease phenotype is conspicuously different. The first paragraph of the introduction section has been revised to include this valuable point.
In order to accommodate clearer interpretation, the figure has been replaced by a simpler version and the information on DAPC deficiencies in mdx versus DMD is now contained within the new Table 1.
Figure 2 shows presence of NFAT5 protein by western blotting in one of four muscles of DMD ,NFAT5 could play a role in permanent regeneration but extracellular preoduction of matrix would inhibit this and could interfere as a possible binding site of glucocorticoid receptor(GR), differences in GR polymorphisms were found by Bonifati et al in steroid response in DMD children ,that is different according to his polymorphisms (J Neurol Neurosurg Psychiatry2006 Oct;77(10):1177-9)
We agree with the reviewer that these polymorphisms could explain resistance to GCs. Differences in GCs sensitivity are a phenomenon also described in DMD by Bonifati et al. If GR resistance is the underlying mechanism, that could explain differences in sensitivity over time and GR interacting with NFAT5, and an impact on NFAT5 expression could be expected in DMD fibroblasts. Indeed, if GR levels diminish over time, NFAT5 could re-emerge in DMD fibroblasts, thereby helping these cells to regain their fibrotic potential.
Supportive treatment can be first tried in cell models, these therapeutic molecules could be tried in DMD myotubes or mdx myotubes(according to Terrill,2020 taurine supplementation decreases their necrosis), subsequently in animal models both in mdx and dog model, may be ultimately tested in DMD/ BMD patients,although carrying out trials is nowadays difficulty both for the relative rarity of dystrophinopaties and family expectations, are also worth trials in BMD (Acta Myol. 2019),in order to counteract both cardiomyopathy and osteoporosis.
This remark is now addressed in the conclusion section. Of note, the study of Terrill et al. in 2020 is referred to in section 2.5.1 where the beneficial effects of taurine on histopathological features such as necrosis is discussed.
The review presents a series of possible drugs to be tested as supplementary treatment in DMD, some were previously subjected to trials and their differential effects could be presented.
We thank the reviewer for suggesting clearer integration of clinical trials. An overview of the osmolytes involved in DMD and the clinical trials is now given in the new Table 2. Unfortunately, the results of these trials have often not been revealed.
Reviewer 2 Report
The manuscript of Herbelet et al., reviews the role of organic osmolytes pathway in the physiopathology of the Duchenne`s muscular dystrophy. After a revision of the role of the protein kinase A/ mitogen-activated protein kinase/ nuclear factor of activated T-cells 5/ organic osmolytes (PKA-p38MAPK-NFAT5-organic osmolytes) pathway in muscle cells regeneration after injury and/or genetic mutation. The authors discuss the possible therapeutic effect of specific (i.e.) or general () pharmacological substances acting on the PKA-p38MAPK-NFAT5 pathway. Overall, the paper is well written and figure 1 is very useful to understand the organic osmotic pathway. In my opinion the paper fill the expectations of a review, and therefore it is suitable for ubication in the present form.
Author Response
We thank the reviewer warmly for the appreciation of our work.
Reviewer 3 Report
In the review entitled "The PKA-p38MAPK-NFAT5-organic osmolytes pathway in Duchenne muscular dystrophy: From essential player in osmotic homeostasis, inflammation and skeletal muscle regeneration to therapeutic target" the authors describe the involvement of PKA-p38MAPK-NFAT5-organic osmolytes pathway in Duchenne muscular dystrophy (DMD) pathophysiology, and its modulation as a potential combinatorial treatment together with genetic therapy to DMD. Although the analysis of the above mentioned pathway could be of interest, the review appears overall confusing about the modulation of this pathway, and a clear description of the signalling sequence PKA-p38MAPK-NFAT5-organic osmolytes in DMD muscles is lacking.
Major points
- In the second paragraph of the Introduction the authors link the activation of NF-κB to TLRs, however TLRs are not the only receptors involved in the NF-κB activation in DMD. Moreover, other pathways have been involved recently in the DMD pathology.
- The physiological role of PKA-p38MAPK-NFAT5-organic osmolytes pathway in skeletal muscles should be described in the Introduction, and the link between this pathway and the absence of dystrophin in DMD should be stressed.
- In the paragraph 2.2. p38 mitogen-activated protein kinase, the role of p38 in muscle regeneration and in aging or other atrophying conditions should be clarified.
- In the paragraph 2.3. Glycogen synthase kinase 3 beta, the authors should introduce GSK-3β before in the text. It is not clear the link between GSK-3β and the pathway analysed.
- The authors should insert a table resuming the role of organic osmolytes in skeletal muscle physiology and in DMD.
Minor points
- Figure 1 is referred to a DMD patient; please remove the differential expressions of DAPC components and PKA-p38MAPK-NFAT5-organic osmolyte pathway members in DMD patients and mdx mice. The authors could insert this aspect in a dedicated table.
- First line of Introduction: please replace "muscular" with “muscle”.
- Briefly describe the DMD pathology in the Introduction.
Author Response
Our sincerest gratitute to the reviewer for the constructive remarks, which we address point-by-point hereunder:
In the review entitled "The PKA-p38MAPK-NFAT5-organic osmolytes pathway in Duchenne muscular dystrophy: From essential player in osmotic homeostasis, inflammation and skeletal muscle regeneration to therapeutic target" the authors describe the involvement of PKA-p38MAPK-NFAT5-organic osmolytes pathway in Duchenne muscular dystrophy (DMD) pathophysiology, and its modulation as a potential combinatorial treatment together with genetic therapy to DMD. Although the analysis of the above mentioned pathway could be of interest, the review appears overall confusing about the modulation of this pathway, and a clear description of the signalling sequence PKA-p38MAPK-NFAT5-organic osmolytes in DMD muscles is lacking.
The text has been revised to better explain the sequence of events that connect signaling pathway members.
Major points
- In the second paragraph of the Introduction the authors link the activation of NF-κB to TLRs, however TLRs are not the only receptors involved in the NF-κB activation in DMD. Moreover, other pathways have been involved recently in the DMD pathology.
We agree with this pertinent remark, and have added cytokine-mediated signaling driven by TNF-α/IL-1β/IL-6 to the introduction section.
- The physiological role of PKA-p38MAPK-NFAT5-organic osmolytes pathway in skeletal muscles should be described in the Introduction, and the link between this pathway and the absence of dystrophin in DMD should be stressed.
Explanation of the involvement of osmolytes in normal muscle cell physiology, and the repercussions for the affected muscle fibers of DMD, has been added to the last paragraph of the introduction section.
- In the paragraph 2.2. p38 mitogen-activated protein kinase, the role of p38 in muscle regeneration and in aging or other atrophying conditions should be clarified.
The requested clarifications have been added to the revised text, referring to reference 48.
- In the paragraph 2.3. Glycogen synthase kinase 3 beta, the authors should introduce GSK-3β before in the text. It is not clear the link between GSK-3β and the pathway analysed.
The involvement of GSK-3 in healthy muscle and in DMD has now been properly introduced in section 2.3.
- The authors should insert a table resuming the role of organic osmolytes in skeletal muscle physiology and in DMD.
Table 2 has been inserted and provides a short overview of the role of organic osmolytes and its expression in DMD patients and/or mdx mice.
Minor points
- Figure 1 is referred to a DMD patient; please remove the differential expressions of DAPC components and PKA-p38MAPK-NFAT5-organic osmolyte pathway members in DMD patients and mdx mice. The authors could insert this aspect in a dedicated table.
In order to accommodate clearer interpretation, the figure has been replaced by a simpler version and the information on DAPC deficiencies in mdx versus DMD is now contained within the new Table 1.
- First line of Introduction: please replace "muscular" with “muscle”.
Muscular was removed from the text.
- Briefly describe the DMD pathology in the Introduction.
The first paragraph of the introduction now contains the requested information on Duchenne disease pathology.
Round 2
Reviewer 1 Report
None